# Magnetic Stimulation as a Therapeutic Approach for Brain Modulation and Repair: Underlying Molecular and Cellular Mechanisms

**DOI:** 10.3390/ijms242216456

**Published:** 2023-11-17

**Authors:** Tom Dufor, Ann M. Lohof, Rachel M. Sherrard

**Affiliations:** 1Department of Cell and Developmental Biology, University College London, London WC1E 6BT, UK; 2Sorbonne Université and CNRS, UMR8256 Biological Adaptation and Ageing, 75005 Paris, France; ann.lohof@sorbonne-universite.fr

**Keywords:** rTMS, LI-rTMS, magnetic stimulation, neuromodulation, non-invasive brain stimulation, magnetoreceptor

## Abstract

Neurological and psychiatric diseases generally have no cure, so innovative non-pharmacological treatments, including non-invasive brain stimulation, are interesting therapeutic tools as they aim to trigger intrinsic neural repair mechanisms. A common brain stimulation technique involves the application of pulsed magnetic fields to affected brain regions. However, investigations of magnetic brain stimulation are complicated by the use of many different stimulation parameters. Magnetic brain stimulation is usually divided into two poorly connected approaches: (1) clinically used high-intensity stimulation (0.5–2 Tesla, T) and (2) experimental or epidemiologically studied low-intensity stimulation (μT–mT). Human tests of both approaches are reported to have beneficial outcomes, but the underlying biology is unclear, and thus optimal stimulation parameters remain ill defined. Here, we aim to bring together what is known about the biology of magnetic brain stimulation from human, animal, and in vitro studies. We identify the common effects of different stimulation protocols; show how different types of pulsed magnetic fields interact with nervous tissue; and describe cellular mechanisms underlying their effects—from intracellular signalling cascades, through synaptic plasticity and the modulation of network activity, to long-term structural changes in neural circuits. Recent advances in magneto-biology show clear mechanisms that may explain low-intensity stimulation effects in the brain. With its large breadth of stimulation parameters, not available to high-intensity stimulation, low-intensity focal magnetic stimulation becomes a potentially powerful treatment tool for human application.

## 1. Introduction: Why Use Magnetic Fields to Stimulate the Brain?

The human brain is extraordinarily complex, and the mechanisms underlying many of its functions remain imperfectly understood. Therefore, repairing neuronal dysfunction or damage remains one of the major challenges in biomedical science, and neural dysfunction continues to seriously impair the quality of life of affected people. Most neurological diseases can only be treated with behavioural remediation, and/or pharmacological treatments, which often have associated side effects. Non-pharmacological, non-invasive brain stimulation (NIBS) is increasingly tested as a potential therapy in neurology and psychiatry, with the aim of triggering intrinsic neuromodulatory and brain-repair molecular and cellular mechanisms.

One approach to extrinsic brain stimulation is the application of pulsed electromagnetic fields to the human brain [1,2] (Figure 1). Because this stimulation was first observed and then used in humans, fundamental studies on stimulation parameters and cellular mechanisms were not initially undertaken. As a consequence, there are no universally defined stimulation protocols, and therefore functional outcomes of magnetic brain stimulation are variable between subjects and studies, and underlying molecular mechanisms remain ill defined [3,4,5,6]. Information about these underlying cellular mechanisms is necessary if we aim to identify the most appropriate stimulation protocols for different brain areas; at different life stages (e.g., child, adult, or aged); and in different pathological conditions, e.g., developmental disorder, injury, or neurodegenerative disease [7,8].

There are two major approaches to NIBS (Figure 2). The first uses trains of high-intensity magnetic pulses (0.5–2 Tesla, T), known as repetitive transcranial magnetic stimulation (rTMS), to induce focal neuronal activation (Figure 2A), which temporarily modifies cortical excitability (the readiness for neurons to be activated), cognition, and behaviour [4,6]. It is assumed that mechanisms of activity-dependent plasticity and/or the entrainment of the waves of neuron activity (neural oscillations) modulate activity between brain regions [6,11]. However, because of the high voltages required to generate these magnetic fields, the stimulation frequency is limited by the time needed to recharge the coil, so in practice, 20 Hz is almost the maximum frequency [4,6]. There are significant other limitations to high-intensity stimulation (e.g., risk of seizures, requirement for a hospital setting, etc.) [4,6]. A second, alternative approach (Figure 2B,C) involves the application of low-intensity transcranial current stimulation or weak pulsed magnetic fields [3] to modify neuronal responses (neuromodulation) throughout the brain, which also induces similar short-term effects on cortical excitability and behaviour but without triggering direct neuronal firing [3,12]. However, the clinical outcomes of diffuse neuromodulation are disappointing [13]. Thus, neither of these approaches provides flexibility in stimulation protocols and the targeting of a specific brain region necessary to treat dysfunctional neural circuits.

Finally, we have developed focal low-intensity magnetic stimulation (LI-rTMS; [15]), which benefits from the wide range of stimulation parameters of weak magnetic fields as well as the focal nature of high-intensity rTMS (Figure 3). Acute effects of 10 min LI-rTMS show changes to the resting state network (regional activity at rest, shown with functional brain imaging (fMRI)) [16] and the modulation of visual evoked responses to concurrent visual stimuli [17]. Also, chronic (2-week) LI-rTMS can induce neural circuit plasticity, stimulating collateral axon outgrowth and reinnervation to denervated cerebellar Purkinje cells in a stimulus-specific manner [18]. LI-rTMS can also remove abnormal neural connections. For example, ephrin-A2A5^−/−^ knockout mice lack key axon guidance cues and, as a result, have disrupted topography in visual pathways and abnormal visual tracking behaviour [19]. Two-week LI-rTMS (10 mT; BHFS; 10 min/day) decreased the number of abnormal projections in subcortical [15] and cortical visual circuits [20], corrected the defective visual tracking [15], and improved accuracy in a visual learning task [21]. Importantly, the effects of this type of stimulation are frequency/rhythm-dependent [18,22], similar to the effects of high-intensity rTMS. Moreover, magnetic stimulation at these lower neuromodulatory intensities has a wider range of possible stimulation parameters [23,24].

The availability of these different magnetic stimulation techniques reinforces the need to understand the molecular underpinnings of the effects of magnetic fields on brain cells at high or low intensity and with different stimulation parameters, in order to develop new protocols for different pathologies. This is particularly true for low-intensity stimulation because, in the absence of overt neuronal firing, there is concern that the observed effects may be a placebo effect rather than real. Here, we review some basic biology combined with animal and clinical studies to compare what is known about mechanisms underlying the effects of high- or low-intensity pulsed magnetic field stimulation on neurons. We consider parameters such as intensity, frequency, and pattern, in order to better understand the potential advantages and disadvantages of different stimulation approaches for human neurological and psychiatric disease. Intriguingly, low- and high-intensity magnetic stimulation techniques appear to share many cellular mechanisms. However, we will not address the vast literature on clinically applied TMS, except to illustrate some of the likely underlying mechanisms.

## 2. Magnetic Brain Stimulation Parameters Determine Outcomes

Transcranial magnetic stimulation is based on Faraday’s principles of electromagnetic induction. A stimulating coil is held over a subject’s head, and a brief pulsed (time-varying) primary current flows through the coil and generates a changing (time-varying) magnetic field perpendicular to the primary electric current. The magnetic field passes through the subject’s scalp and skull with negligible attenuation and induces a secondary electric current, parallel to but in the opposite direction from the coil’s current, and thus stimulates the brain [25] (Figure 1). The magnetic field decreases rapidly with distance from the coil; thus, the cortex and subcortical white matter are assumed to be the principal neural elements stimulated.

The outcomes of rTMS are influenced by a large number of stimulation parameters, as well as characteristics of individual subjects’ brains (Figure 4).

### 2.1. Types of Coil

Coil designs available for clinical TMS are usually either classic round coils or figure-of-eight coils [26,27]. These coils produce magnetic fields of different shapes, which will determine the area with the highest intensity stimulation (“hotspot”), the total area stimulated, and the stimulation depth [28,29].

### 2.2. Stimulation Intensity

In current clinical practice, rTMS aims to induce neuronal firing in the targeted cortical region. The intensity of rTMS applied is in the range of 0.5–2 T and is usually determined with the cortical resting motor threshold (RMT) [6]. This is defined as the minimal intensity at which TMS over M1 reliably induces enough action potential firing to produce a detectable electromyographic motor evoked potential (MEP) around 100 µV or a visible contraction in the target muscle [30]. MEP amplitude is therefore used as an indirect measure of cortical excitability and increases with increasing stimulation intensity [31]. In contrast, low-field magnetic stimulation (LFMS) parameters are usually in the µT-mT range [3]. Although MEPs have been used to measure the effect of LFMS [32], the more commonly used read-out is electroencephalography (EEG) brain oscillations. In a comparison between the effects of 1.5 and 10 Hz stimulation at 20–40 μT, 10 Hz induced a greater increase in EEG power (oscillation amplitude). Notably, stimulation at specific electromagnetic field frequencies influences EEG power in the same frequency range [33]. Thus, alpha and beta EEG bands are entrained by corresponding LFMS frequencies [34], which is consistent with EEG entrainment by high-intensity rTMS [11,35].

Comparisons between rTMS studies are complicated by the fact that applied stimulus strength depends on (a) intrinsic differences in cortical excitability that alter the RMT [36] and (b) the device used, since stimulation intensity is usually expressed as the percentage of maximal stimulator output inducing RMT rather than as a voltage or magnetic field in Tesla/Gauss [31,37]. Moreover, the target brain area will necessarily be surrounded by tissue exposed to lower-intensity magnetic stimulation, which will also induce its own effects on the associated neurons and their circuits.

### 2.3. Frequency and Pattern of Magnetic Stimulation Pulses

Irrespective of stimulation intensity, whether in the Tesla or milliTesla range, the temporal spacing between repeated magnetic stimuli (rTMS frequency) is the most widely studied parameter of magnetic stimulation because of the known effects on cortical excitability. For clinical high-intensity rTMS, high frequencies (≥5 Hz) increase cortical excitability, and low frequencies (≤1 Hz) decrease it [36,37,38]. Theta burst stimulation (TBS) is a complex pattern (three-pulse 50 Hz bursts repeated at 5 Hz) that can be delivered as continuous (cTBS) or intermittent (iTBS) trains; cTBS exerts inhibitory effects, and iTBS exerts excitatory effects on cortical excitability [38]. Both iTBS and cTBS induce the sustained modulation of human cortical excitability, even when delivered at a lower intensity and shorter duration than classical high- and low-frequency rTMS [38,39,40]. Finally, when stimulation intensity is low (e.g., mT), pulses can be delivered at a wide range of frequencies, which are generally within a 10–100 Hz range [3].

All rTMS protocols, whether at high or low intensity, simultaneously modulate multiple cortical circuits [39,41], inducing mixed inhibitory and excitatory effects [42]. Indeed, the total number of pulses delivered, and the interval between the blocks of stimulation, result in the recruitment of different cortical interneuron networks [43] to increase or decrease the effects of TBS on cortical excitability [44,45], and this may explain some inter-individual variability [43].

## 3. Potential Cellular Mechanisms Mediating the Effects of Magnetic Stimulation

An understanding of the mechanisms underlying rTMS will help to fine-tune protocols in order to optimally activate appropriate neural repair mechanisms, depending on the individual subject, the neural network targeted, and the pathology involved. Studies using different stimulation intensities show that many cellular mechanisms are shared between high and low intensity.

### 3.1. Synaptic Plasticity

TMS simultaneously activates a large number of neurons in local circuits; thus, a mixture of pre-synaptic and post-synaptic neurons are activated [46]. It modulates cortical excitability beyond the simulation period [4] as indicated by EEG [47,48], regional cerebral blood flow [49,50], and blood oxygen level-dependent (BOLD) activation patterns [51]. All of these changes suggest synaptic and network plasticity. This plasticity induced by rTMS is influenced by the prior activation of the neural circuit. This is in accordance with the concept of metaplasticity, which facilitates or inhibits plastic change, stabilises synapses, or adjusts cellular activity in the context of prior activity [52]. An important component is homeostatic synaptic plasticity, which adjusts synaptic strength to maintain stable neural activity. Several rTMS studies have shown that previous activity in the stimulated network influences the observed outcomes [53], i.e., prior stimulation “primes” the circuit and influences the subsequent stimulation effects [54,55].

Magnetic stimulation-induced plasticity can be either excitatory or inhibitory depending on which neuronal population is most affected by the stimulation protocol. In vitro studies of high-intensity repetitive magnetic stimulation (rMS; i.e., the magnetic field is not transcranial) on organotypic rodent hippocampal slice cultures have delivered direct evidence of LTP induced by high-frequency rMS (10 Hz) [56,57,58]. This induced LTP is durable (2–6 h) and occurs at excitatory synapses on proximal dendrites, in association with the remodelling of small dendritic spines and increased receptor clusters; it also requires voltage-gated ion channels, NMDA receptors, and calcium [56,58], similar to classical electrophysiological LTP. Moreover, rTMS can also strongly inhibit cortical networks [25], through modulating inhibitory interneuron activity [46,59,60].

Different stimulation patterns affect distinct interneuron populations appropriate to their observed effects on cortical excitability (e.g., MEP). According to the pattern applied, rTMS modulates the inhibitory interneuron expression of the immediate early genes c-fos and zif268 [46,61,62], GABA-synthesising enzymes GAD65 and GAD67 [46,60,63], and calcium-binding proteins [59,63,64]. Stimulation patterns that increase cortical excitability, such as 10 Hz and iTBS stimulation, not only induce Ca^2+^-dependent signalling [56,58,59,65,66] but also depress inhibitory circuits by reducing parvalbumin (PV) expression in fast-spiking interneurons (FSIs) [62,67,68,69] and destabilising GABA receptors to reduce GABAergic synaptic strength [66]. In contrast, cTBS and 1 Hz (“inhibitory” protocols) predominantly alter calbindin (CB) expression [67,68]. This interneuron sensitivity to TMS develops with FSI maturation between P30 and 40 in mice (equivalent to human adolescence) when perineuronal nets stabilise the cortical synaptic network [64,70].

Taken together with the effects at excitatory synapses, these fundamental experiments reveal the complex biology of brain magnetic stimulation; for example, an excitatory effect may include increased synaptic plasticity in excitatory circuits along with reduced inhibitory activity.

### 3.2. Acute Effects on Neuronal Properties

#### 3.2.1. Membrane Potential and Spontaneous Activity

Magnetic stimulation can change the membrane potential of targeted neurons either directly, or indirectly via interneurons [4,71]. Both whole-cell recordings and voltage-sensitive dyes show that magnetic pulses initiate a transient current into neurons through voltage-gated sodium channels [72,73]. The electric field induced by high-intensity rTMS (0.5–2 T) is known to be sufficient to produce neuronal activation, i.e., action potentials (see above). Although the electric field induced by weak rTMS (~10 mT) was assumed to be below the neuronal activation threshold, some stimulation pulses (~25%), are nonetheless followed by action potentials when weak rTMS is applied [71]. An increase in the intracellular calcium following excitatory protocols [72] progressively increases activity in a large population of neurons that weakens inhibitory action and stimulates excitatory circuits through NMDA receptors [65,74]. Depending on the stimulation pattern, weak rTMS modulates neuronal excitability and changes the membrane potential [71].

These rapid changes to the membrane potential and neuronal excitability may explain how high-frequency rTMS reverses age-related reduction in neuronal excitability and cognitive function [75], which are partly due to a hyperpolarised resting membrane potential [76] and greater after hyperpolarisation [77,78]. Also, rTMS modifies voltage-gated Ca^2+^ channel (VGCC) activity [75], which improves synaptic plasticity in aged neurons [79].

#### 3.2.2. Activation of Cryptochrome Magnetoreceptors

The knowledge that low-intensity magnetic fields can generate action potentials in neurons is very recent [71]. In contrast, the basic biology of magnetic fields has been studied in various systems, addressing the direct interaction between external magnetic fields and the biological system. However, this knowledge has not previously been integrated into the domains of human or animal rTMS/LFMS.

Interaction between magnetic fields and biological tissues—magnetoreception—allows, amongst other functions, for orientation to the ambient geomagnetic field [80,81,82] and the regulation of circadian rhythms [83]. This phenomenon involves cryptochrome (CRY; Figure 5), which is widely conserved across species [84,85]. CRY transduces electromagnetic signals through the activation of its FAD cofactor followed by electron transfer from a conserved triad of tryptophan residues [86,87]. This stimulation induces conformational changes in the cryptochrome protein, thus removing its inhibition of Clock/Arntl transcriptional activity [88], and generates a radical pair [80,89], which in turn generates reactive oxygen species (ROS; Figure 5) [90,91]. Both ROS and changes to gene expression are induced by rTMS [92,93], and the increase in ROS requires CRY [93]. Moreover, the olivocerebellar reinnervation induced by LI-rTMS after a lesion is abolished in CRY double-knockout mice [18].

### 3.3. Intracellular Cascades Modified by Magnetic Fields

#### 3.3.1. ROS Production and Regulation of Oxidative Stress

ROS production is considered an early response to LFMS exposure [92,97,98]. ROS were originally characterised as harmful to cells, but accumulating evidence shows that at physiological concentrations, they participate in signal transduction, Ca^2+^ release from intracellular stores, and the fine-tuning of cellular signalling [99].

In normal cells, magnetic stimulation effects on ROS production show small amplitude changes (30–60%) in either direction; this “low-level” oxidative modulation is compatible with non-toxic and even protective effects of magnetic field exposure [92,98]. Moreover, when neurons are already stressed, e.g., after ischaemia, the small modulation of ROS via LFMS activates reparative antioxidants [97] and heat shock protein pathways [100] to reduce oxidative damage.

#### 3.3.2. Intracellular Ca^2+^ Concentration and Downstream Signalling

Signalling cascades underlying magnetic stimulation often involve intracellular calcium [101,102]. Induced action potentials increase intraneuronal calcium through voltage-gated calcium channels (VGCCs), and synaptic activity can increase calcium influx through NMDA receptors, which may partially explain changes to neural activity [72,103,104,105]. Magnetic stimulation has also been shown to increase calcium release from intracellular stores [22,101] through the action of ROS [106].

Calcium activates various enzymes, and at least two of these, namely calmodulin-dependent kinase II (CAMKII) and calcineurin, regulate the initiation and maintenance of LTP and LTD, respectively [107]. Both CAMKII and calcineurin are activated by the same Ca^2+^/calmodulin complex [108]. Calmodulin very rapidly buffers calcium, activating and inactivating CAMKII/calcineurin in parallel with neuronal firing [109], thus providing a potential mechanism to mediate the effects of pulsed magnetic fields. Indeed, in vitro, 10 Hz rTMS upregulates CAMKII [110], and calcineurin underlies post-stimulation changes in inhibitory synaptic responses [66]. Ca^2+^ also regulates nitric oxide through calmodulin binding and neuronal nitric oxide synthase activation [111], which regulates synaptic plasticity and network function [15,20,65,112,113].

These studies suggest that calcium dynamics are involved at the early stage of rTMS-induced plasticity. Calcium signalling due to rTMS-induced neuronal activity increases calcium-dependent kinase cascades, leading to immediate-early gene (IEG) expression. This in turn activates the expression of target genes, leading to long-term functional and structural modifications to neurons [114] (Figure 6).

#### 3.3.3. Immediate-Early Gene Expression

Immediate-early genes (IEGs) are rapidly induced, transiently expressed genes [115,116]. Changes in their expression were first associated with rTMS when 1 Hz stimulation to human-derived neuron-like cells increased intracellular cAMP; CREB phosphorylation; and the downstream expression of neuronal plasticity associated IEGs, c-fos, and zif268 [117].

Robust evidence for increases in IEG expression via magnetic stimulation now exists. Notably, c-fos expression was increased in cultured cortical neurons via 1 Hz and 10 Hz rTMS [22]. In addition, c-fos upregulation was specifically strong in those neurons undergoing biological responses to LI-rTMS (e.g., reinnervation [18]). In vivo, Zif268 expression was increased in almost all cortical areas after iTBS but only in the primary motor and sensory cortices after 10 Hz rTMS [61]. A recent study also showed that c-fos increases can be found in connected cortical regions [62].

These gene expression changes occur in a time course with distinct phases. In the awake rat [118], the administration of 3 min iTBS to the cortex increased c-fos and GAD65 expression over 20 min, indicating the strong activation of excitatory and inhibitory neurons. This was followed by a phase (20–80 min) of reduced inhibitory activity, as reflected by decreased GAD67, PV, and CB expression. Then, finally (after 160 min), c-fos expression disappeared, suggesting the end of the iTBS-induced cortical facilitation [118]. As the number of iTBS stimulation blocks, or intervals between them, were changed, so did the profiles of gene expression [46,60], consistent with the effects of different TBS stimulation parameters on human cortical excitability [44,45].

These regional and stimulation-specific changes to gene expression introduce a fundamental aspect to rTMS that is not yet applied in clinical practise: each brain region, with its individual neuron populations and microcircuity, will respond differently to any given stimulation protocol.

### 3.4. Long-Lasting Brain Plasticity

An important component of magnetic stimulation’s therapeutic potential is that it modulates cortical excitability beyond the simulation period [4], indicating the induction of mechanisms for long-lasting plasticity, such as growth factor expression, alterations to dendritic spines, and neurogenesis.

#### 3.4.1. Brain-Derived Neurotrophic Factor and Other Gene-Expression Changes

The brain-derived neurotrophic factor (BDNF) is probably the most-studied neurotrophin in the rTMS field. The BDNF is widely expressed in the CNS and is essential for normal brain development and function [119] including the induction of LTP through Ca^2+^ and IEG expression [120,121]. Its link to rTMS is shown by reduced BDNF in CRY-deficient mice [122] and poor responsiveness to TMS in BDNF Val66Met allele carriers [123], as well as increased serum BDNF after long-term rTMS [124]. However, studies of increased BDNF after the administration of acute rTMS to depressed patients yield inconsistent results [125,126,127].

Animal studies generally support this view, but they also reveal the greater complexity of BDNF regulation. In the acute phases of stimulation, changes to BDNF expression are frequency- and state-dependent. High (20 Hz), but not low (1 Hz), frequency rTMS at high intensity increased BDNF expression in awake rats but reduced the BDNF in anesthetised rats, highlighting the importance of neural activation state during stimulation [5,128]. After repeated rTMS sessions over several weeks, BDNF expression increased in hippocampal neuron cultures [129], in the brains of adult and aged rodents [116,130], and in those with stroke or vascular dementia [131,132]. In addition, at lower stimulation intensity, the BDNF is increased in the tectum and cortex in parallel with LI-rTMS-induced visual circuit reorganisation [15,20].

Moreover, changes in the expression of other genes, including apoptosis and neurite outgrowth genes, were observed following LI-rMS in vitro [22], consistent with neuron survival and neurite branching. Finally, the upregulation of axon outgrowth and synaptogenesis genes are induced only through LI-rTMS frequencies, inducing olivocerebellar reinnervation [18].

#### 3.4.2. Dendritic Spine Plasticity

Another long-term correlate of synaptic plasticity affected by magnetic stimulation is the modification of dendritic spines, whose morphology changes with synaptic activity. Consistent with this, 10 Hz rTMS to hippocampal slices altered dendritic spine size, in agreement with its induction of LTP [58], and high-frequency LI-rTMS increased spines in Purkinje neurons [133].

#### 3.4.3. Neurogenesis

Neurogenesis in the subventricular zone and the hippocampal dentate gyrus subgranular zone is an important part of neural circuit plasticity [134]. High-frequency stimulation (15–50 Hz) over a few weeks increases newborn neurons in these zones [135,136,137], including promoting their differentiation [104,105] and survival [103,138]. How rTMS induces neurogenesis is unknown, although BDNF upregulation and/or excitatory neuronal activity are both likely mechanisms [136,137].

These data indicate that magnetic brain stimulation induces a series of effects on the brain from modulating neuron network excitability to altering neuronal intracellular signalling and gene expression. These intrinsic modifications, e.g., synaptic plasticity or BDNF upregulation, are known to have positive effects on injured and dysfunctional brain systems and support the applicability of rTMS to treat brain pathology. Moreover, many processes underlying rTMS/LI-rTMS are essentially similar whether the stimulation is at high intensity or low intensity. Thus, while there is initial evidence that magnetic stimulation can repair abnormal and damaged neural circuits, appropriate stimulation parameters remain to be identified for each lesion type and region affected [18].

## 4. Relevance of Magnetic Stimulation to Neuropsychiatric Treatment

Extensive research has already been carried out on the application of magnetic stimulation to treat human neuropathology [3,4,5,6,10] and the combined information from areas as diverse as plant and insect biology, in vitro models, animal and human studies using both high- and low-intensity stimulation show that common processes are involved. These include Ca^2+^ signalling, IEG expression, BDNF upregulation, altered gene expression, and, possibly, the activation of magnetoreceptors. Accordingly, because these cellular processes are common to both types of magnetic stimulation, it seems likely that rTMS and LFMS/LI-rTMS represent a continuum of the same treatment strategy, and thus information from both fields should be combined to optimise the understanding, and therefore potential application, of magnetic brain stimulation. In this context, we suggest that cortical and subcortical low-intensity stimulation that surrounds an rTMS “hotspot” [28] contributes to the behavioural effects of the high-intensity stimulus.

In addition, identifying cellular processes modified via magnetic stimulation—such as Ca^2+^ signalling and ROS production, both of which are toxic in high concentrations—can also explain how rTMS has the potential to induce adverse effects. Not only does longer duration and higher-intensity stimulation provide no benefit for psychiatry patients [139] and kill neurons in mice [129], but prolonged environmental exposure to weak magnetic fields has also been associated with negative symptoms and even neurodegenerative disease [140,141]. Therefore, rising concentrations of such molecules as nitric oxide or ROS may provide suitable biomarkers for defining maximum safe stimulation protocols.

Lastly, since rTMS is already used clinically, if high- and low-intensity magnetic stimulation processes form a continuum of neurostimulation activating common mechanisms, is there a place for low-intensity magnetic stimulation beyond anecdotal interest? We think so, given the fact that even focal LI-rTMS is sufficient to induce marked neurobiological effects and that some of its mechanisms—through modified neuronal activity and magnetoreceptor activation—have been identified. The greater range of stimulation parameters open to LI-rTMS, and consequently more refined protocols for different pathologies and different brain areas, reinforce suggestions that it is a promising therapeutic tool [24] beyond what has already been shown for depression [5,14,142,143]. Furthermore, using low-intensity magnetic fields would avoid the need for high-voltage electric sources and treatment restrictions in the hospital setting. However, despite what we presented in this review, a systematic investigation of the effects of precisely defined stimulation parameters on biological processes at the system, circuit, and cellular level is missing. A multidisciplinary approach combining electronics, mathematical modelling, and neuroscience is required to elucidate the most appropriate protocols for different brain regions and different pathologies.

## Figures and Tables

**Figure 1 ijms-24-16456-f001:**
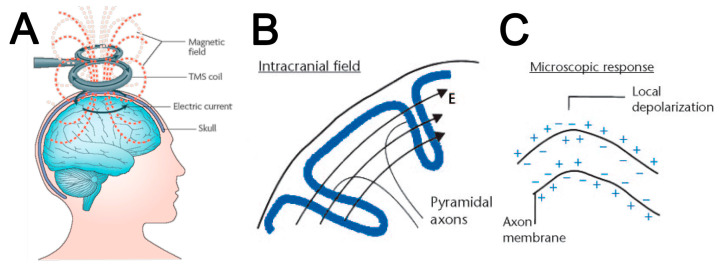
Overview of electromagnetic induction in TMS: (**A**) A brief pulsed current flows through a coil generating a magnetic field, which passes through the skull and induces a secondary current, which stimulates the brain. Image adapted from [9], with permission. (**B**) The electric field (E) within the brain runs through cortical folia essentially perpendicular to the descending pyramidal neuron axons. Image adapted from [10], with permission. (**C**) Microscopically, the secondary current can locally alter the axon membrane potential and therefore activate neuronal firing (image adapted from [10], with permission).

**Figure 2 ijms-24-16456-f002:**
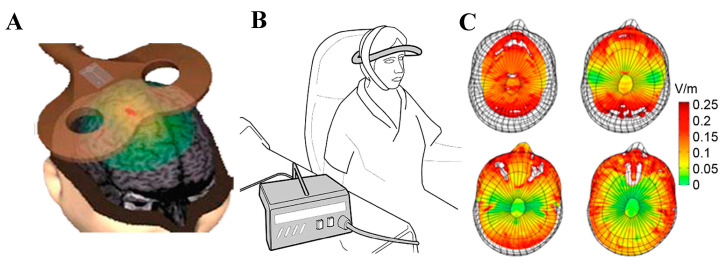
TMS devices and magnetic field propagation in the human brain: (**A**) A figure-of-eight coil held over the human motor cortex to deliver high-intensity TMS with a small central “hotspot” (red) surrounded by a large area of cortex exposed to a much lower field (green) (from http://www.loe.fu-berlin.de/en/dine/labs/tms/index.html, accessed on 2 October 2023). (**B**) Drawing of an LFMS device composed of a custom coil (copper wire 0.2 mm, 1400 turns) on a flexible plastic support (grey ring) strapped around the head with the positive magnetic pole upwards (from [3], with permission). (**C**) Electric field magnitude, induced by low field magnetic stimulation (LFMS) in four transverse slices of the human brain, was calculated by applying the finite element method on a human head model placed inside the LFMS coil (adapted from [14], with permission).

**Figure 3 ijms-24-16456-f003:**
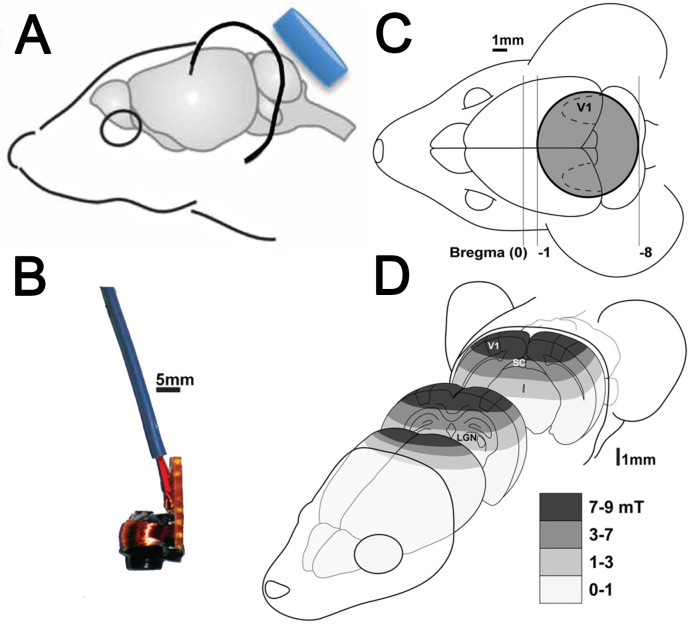
Focal LI-rTMS delivery in mice: (**A**) Custom-built stimulation coil (8 mm diameter; blue) and its relation to the mouse's head for cerebellar stimulation. Modified from [18], with permission. (**B**) Coil with the cover removed to show the wire coil and its support. (**C**) Coil position for bilateral stimulation of the visual cortex (top, grey circle, V1). (**D**) Diagram of magnetic field intensities (in mT) in the mouse head. The magnetic field is relatively focal to the visual cortex; (**B**–**D**) are adapted from [20], with permission.

**Figure 4 ijms-24-16456-f004:**
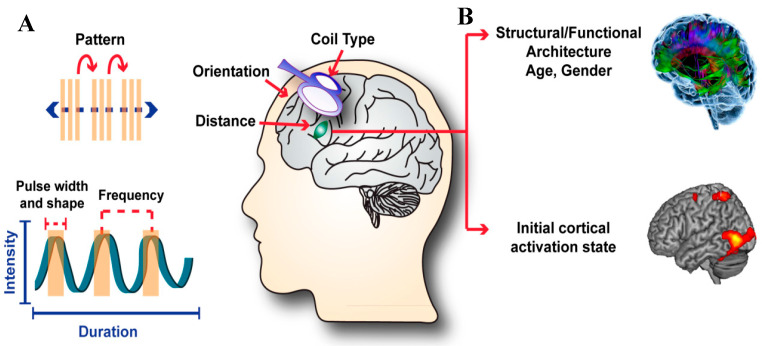
Factors influencing the effects of transcranial magnetic stimulation (TMS): (**A**) TMS pulse parameters that can be tailored for a desired outcome: pattern, frequency, and pulse shape. Coil type, orientation, and distance from the head also alter outcomes. (**B**) A subject’s brain characteristics, including cortical activation state, also influence the outcomes of a given TMS protocol. The different colours in the upper panel represent connections between different brain regions. © Malfeda Loreti and Tom Dufor, used with permission.

**Figure 5 ijms-24-16456-f005:**
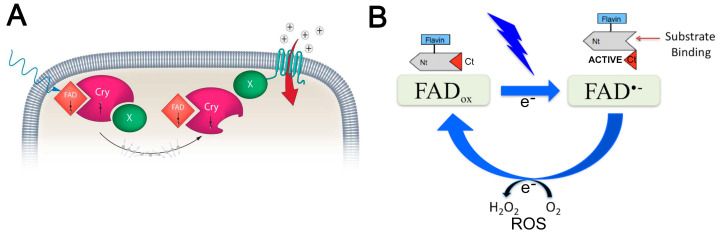
Potential mechanisms of magnetoreception: (**A**) Blue light or magnetic fields (thin blue arrow) induce the formation of radical pairs between CRY and its cofactor FAD. This influences the biochemical or structural properties of CRY, induces a signalling molecule (X), and modulates ion channel permeability. Possibilities for X are Ca^2+^ and ROS; modified from [94], used with permission. (**B**) In the dark, the protein-bound flavin cofactor is oxidised (FAD_ox_). Blue light or magnetic fields (yellow flash) reduce cryptochrome flavin (FAD^•−^), which results in CRY conformational change. Spontaneous reoxidation (FAD_ox_) occurs in the dark and produces ROS [91]. Magnetic fields decrease O_2_^•−^ and increase H_2_O_2_ concentrations and are believed to occur through singlet-triplet modulation of semiquinone flavin (FADH^•^) enzymes and O_2_^•−^ spin-correlated radical pairs [95]; modified from [96], used with permission.

**Figure 6 ijms-24-16456-f006:**
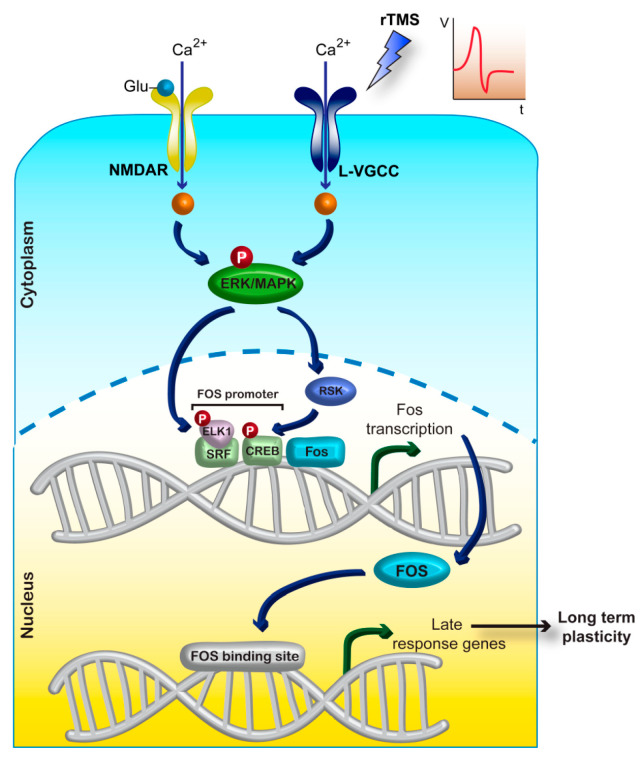
Schema of the potential CREB-Fos pathway activated by rTMS. Calcium influx through NMDA receptors and VGCCs can activate calcium-dependent kinase cascades, leading to CREB activation and thus the expression of immediate early genes (IEGs). © Malfeda Loreti and Tom Dufor, used with permission.

## Data Availability

Not applicable.

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
