# Peer review of "Magnetic Stimulation as a Therapeutic Approach for Brain Modulation and Repair: Underlying Molecular and Cellular Mechanisms"

_ijms, 2023, doi:10.3390/ijms242216456_

Round 1

Reviewer 1 Report

Comments and Suggestions for Authors

Manuscript ijms-2687225 “Magnetic stimulation as a therapeutic approach for brain modulation and repair: underlying molecular and cellular mechanisms” by Tom Dufor, Ann M. Lohof, Rachel M. Sherrard  is an interesting and very informative review in the field of magnetic brain stimulation.

However, the authors did not take into account that the journal they chose for publication was aimed at a very wide readership. Therefore, all abbreviations and special terms used in the text (for example, cortical excitability) must be deciphered and/or briefly explained.

Please, explain how a magnetic stimulation relates to the Ca2+ ion concentration change, if the calcium ions are diamagnetic?

The authors were careless about the requirement to obtain permission to use figures that may be subject to copyright when borrowing them from previously published sources.

For example, in the first three figures, the pictures are directly copied from the cited reviews.

In figure 4, there is no explanation at all, where the images were taken from, and in the publication citation number 25, no such illustration is found. The Figure 5a was not found in the cited article, it was taken from the next web page https://www.chegg.com/homework-help/questions-and-answers/pmagnetoreceptions-light-gray-arrows-labeled-refer-fad-cry-fad -cry-x--enzymatic-reactions--q96542497. In addition, the image used in figure 6 is missing in the cited source at number 114.

Considering the above and taking into account that the manuscript text is rather carelessly formatted, and requires structuring in accordance with the journal format, the presented review can be published only after significant revision.

Author Response

Thank you for this review.  Please see the attached document for our responses.

Reviewer 2 Report

Comments and Suggestions for Authors

In their manuscript entitled “Magnetic stimulation as a therapeutic approach for brain modulation and repair: underlying molecular and cellular mechanisms”, the Authors present a review aiming at presenting an overview regarding the molecular and biological action effects of magnetic stimulation.

This is a very interesting and tamily study. It seems well designed and well written in most of its parts. However, in my opinion, there is a major point to be fixed regarding the objective of the review, that is not totally clear to me.

Indeed, at the end of the introduction section, Authors illustrate some of the characteristics of the low intensity magnetic stimulation they introduced highlighting that the effects of this type of stimiulation depend on several factors and concluding that “This reinforces the need to understand the molecular underpinings of the effects of magnetic fields on brain cells in order to develop new protocols for different pathologies. “

Therefore, one would aspect a review specifically on low intensity magnetic stimulation and not on magnetic stimulation in general.  

Could authors explain better their aims?

Author Response

Thank you for this review.  Please see the attached document for our responses

Round 2

Reviewer 1 Report

Comments and Suggestions for Authors

After the revision, the text of the review became much better perceived. The only remark is that the list of abbreviations used in the article should be placed at the beginning of the text, if possible.

Reviewer 2 Report

Comments and Suggestions for Authors

I thank the Authors for their work and for solving the point I raised in the first round of revisions.